# Electrically programmable magnetic coupling in an Ising network exploiting solid-state ionic gating

Chao Yun[1,2,8], Zhongyu Liang [1,8], Aleš Hrabec [3,4,5], Zhentao Liu[3,4], Mantao Huang [6], Leran Wang[1], Yifei Xiao[7], Yikun Fang[7], Wei Li[7], Wenyun Yang[1], Yanglong Hou [2], Jinbo Yang [1], Laura J. Heyderman [3,4] ✉, Pietro Gambardella [5] ✉ & Zhaochu Luo [1] ✉

Two-dimensional arrays of magnetically coupled nanomagnets provide a mesoscopic platform for exploring collective phenomena as well as realizing a broad range of spintronic devices. In particular, the magnetic coupling plays a critical role in determining the nature of the cooperative behavior and providing new functionalities in nanomagnet-based devices. Here, we create coupled Ising-like nanomagnets in which the coupling between adjacent nanomagnetic regions can be reversibly converted between parallel and antiparallel through solid-state ionic gating. This is achieved with the voltage-control of the magnetic anisotropy in a nanosized region where the symmetric exchange interaction favors parallel alignment and the antisymmetric exchange interaction, namely the Dzyaloshinskii-Moriya interaction, favors antiparallel alignment of the nanomagnet magnetizations. Applying this concept to a two-dimensional lattice, we demonstrate a voltage-controlled phase transition in artificial spin ices. Furthermore, we achieve an addressable control of the individual couplings and realize an electrically programmable Ising network, which opens up new avenues to design nanomagnet-based logic devices and neuromorphic computers.

The ability to electrically manipulate magnetism is crucial for spintronic applications including spin-based data storage and computation devices. The successful switching of the magnetization in nanomagnets by means of spin transfer torques[1], spin-orbit torques[2–4], voltage-controlled magnetic anisotropy (VCMA)[5–8] and magneto-electric coupling[9,10] have led to significant steps towards next-generation low-power and high-speed magnetic memories. However, while nanomagnet-based data storage relies on this magnetization switching, nanomagnet logic requires the engineering of the magnetic coupling that aligns the magnetization of adjacent nanomagnets with a specific relative orientation[11–14]. In addition, with the possibility to construct various two-dimensional (2D) coupled nanomagnetic networks, often referred to as artificial spin ices[15,16], the control of lateral coupling is of particular interest for the investigation of collective phenomena such as magnetic frustration, emergent magnetic monopoles and phase transitions[15–20], as well as for the implementation of

¹State Key Laboratory of Artificial Microstructure and Mesoscopic Physics, School of Physics, Peking University, 100871 Beijing, China. ²School of Materials Science and Engineering, Peking University, 100871 Beijing, China. ³Laboratory for Mesoscopic Systems, Department of Materials, ETH Zurich, 8093 Zurich, Switzerland. ⁴Laboratory for Multiscale Materials Experiments, Paul Scherrer Institute, 5232 Villigen PSI, Switzerland. ⁵Laboratory for Magnetism and Interface Physics, Department of Materials, ETH Zurich, 8093 Zurich, Switzerland. ⁶Department of Materials Science and Engineering, Massachusetts Institute of Technology, Cambridge, MA, USA. ⁷Division of Functional Materials, Central Iron and Steel Research Institute Group, 100081 Beijing, China. ⁸These authors contributed equally: Chao Yun, Zhongyu Liang. ✉e-mail: laura.heyderman@psi.ch; pietro.gambardella@mat.ethz.ch; zhaochu.luo@pku.edu.cn

multiple computation tasks such as Boolean logic operations[11–14,21] and neuromorphic computing[22–26]. Despite the different lateral coupling mechanisms available, including long-range dipolar coupling and nearest-neighbor chiral coupling, these are engineered either through the geometric design[27–30] or by locally tuning the magnetic anisotropy during the fabrication process[19]. As a consequence, the functionality of coupled nanomagnets is determined once fabricated and the subsequent electrical modulation at run-time remains challenging.

Here, we present a mechanism to realize electrically tunable lateral coupling between two adjacent Ising-like nanomagnets by exploiting the VCMA-mediated competition of the symmetric exchange interaction and Dzyaloshinskii-Moriya interaction (DMI) that favor the parallel (P) and antiparallel (AP) alignment of the nanomagnet magnetizations respectively (Fig. 1a). Employing this concept for extended networks, we are able to explore the voltage-controlled phase transition in artificial spin ices and to obtain an electrically programmable nanomagnetic Ising network that can serve as a neuromorphic computing element.

## Results

### Voltage-controlled magnetic coupling

The voltage-controlled nanomagnets are fabricated from a Pt/Co-based multilayer that has a large interfacial DMI[31,32] and tunable perpendicular magnetic anisotropy[33]. As shown in Fig. 1b, the structure comprises two protected regions (in red) with a fixed out-of-plane (OOP) magnetic anisotropy and a 50 nm-wide gated region (in blue) with tunable magnetic anisotropy. In the gated region, the Co layer is exposed to the electrolyte $GdO_x$, constituting a solid-state ionic gate structure[34–36] and thus its magnetic anisotropy can be reversibly

modulated between in-plane (IP) and OOP by applying positive and negative voltages to the top gate electrode, respectively. In contrast, in the protected region where an 8 nm-thick $SiO_2$ layer is inserted between the Co layer and $GdO_x$, the migration of oxygen ions in $GdO_x$ to the top interface of Co is blocked and hence its magnetic anisotropy is protected from voltage modulation. The detailed fabrication process is described in Methods and Supplementary Information S1.

The $GdO_x$ was grown in oxygen-deficient conditions such that, in the as-fabricated device, oxygen ions at the top interface of the Co layer in the gated region are partially absorbed by $GdO_x$. The gated region exhibits IP magnetic anisotropy whereas the surrounding protected regions have OOP magnetic anisotropy, giving an OOP-IP-OOP anisotropy configuration. Moreover, as a result of the interfacial DMI at the Pt/Co interface, the magnetization within the OOP-IP and IP-OOP transition regions twists following a left-handed chirality enforced via chiral coupling[20]. As a result, the two OOP magnetizations in the protected regions are effectively AP coupled (Fig. 1a). In order to obtain the AP ground state, an oscillating and decaying magnetic field is applied perpendicularly to the devices, serving as the demagnetization protocol (Supplementary Information S2). As shown in Fig. 1c (left panel), for an array of coupled elements (Fig. 1b), the magnetization in the two coupled OOP regions exhibits either ↑↓ or ↓↑ alignment. After applying a negative gate voltage $V_G$ of −2.5 V for 90 min, the gated region exhibits OOP anisotropy thanks to the formation of Co-O bonds promoting perpendicular magnetic anisotropy[33]. Due to the collinear alignment in the OOP-OOP-OOP configuration, the influence of DMI is reduced and the symmetric exchange interaction results in ↑↑ and ↓↓ low energy states. In other words, the two OOP magnetizations in the protected regions are P coupled. The conversion from AP to P coupling

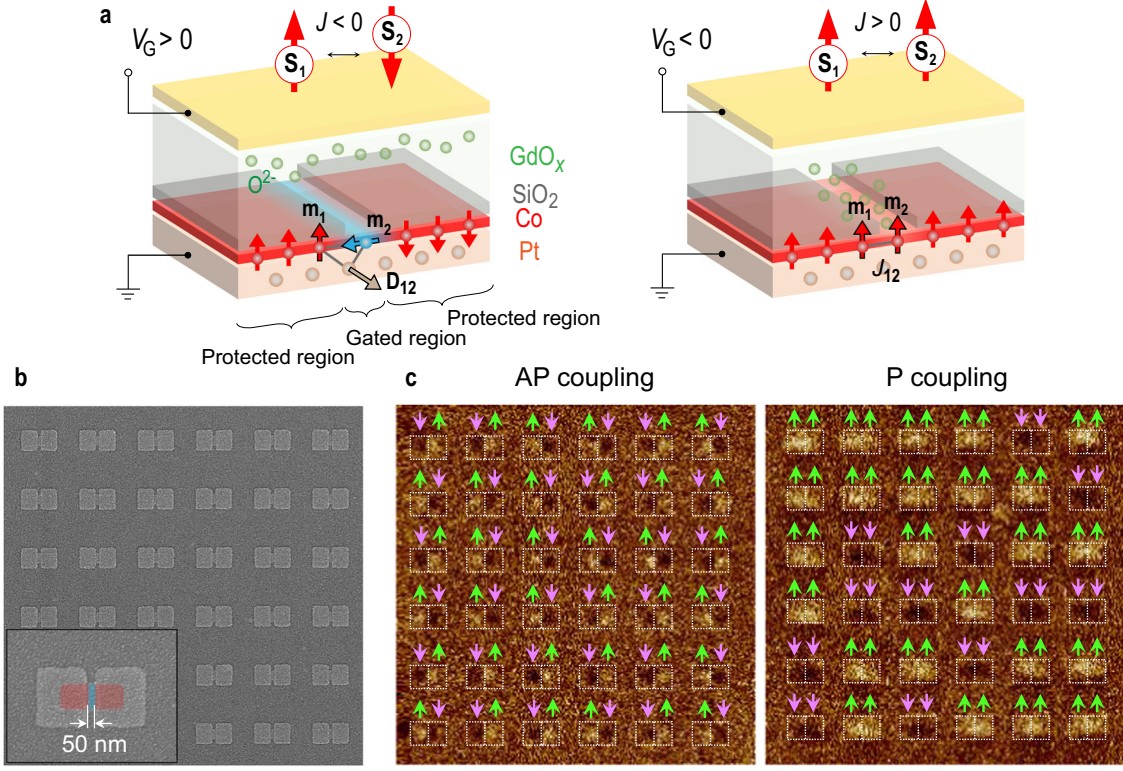

**Fig. 1 | Basic concept of voltage-controlled magnetic coupling. a** Schematics of coupled nanomagnet elements illustrating the principle of AP/P coupling conversion. On application of positive gate voltages, oxygen ions migrate away from the Co interface and the coupling between two protected regions is AP induced by interfacial DMI: $H_{DMI} = -\mathbf{D}_{12} \cdot (\mathbf{m}_1 \times \mathbf{m}_2)$, where $\mathbf{D}_{12}$ is the DM vector, and $\mathbf{m}_1$ and $\mathbf{m}_2$ are two nearest-neighbor magnetic moments. On application of negative gate voltages, oxygen ions migrate to the Co interface and the coupling becomes P due

to the symmetric exchange interaction. **b** Scanning electron microscope (SEM) image of a 6 × 6 array of nanomagnetic structures. As shown in the inset, red and blue colors indicate the protected and gated regions respectively. **c** MFM images of the 6 × 6 array with AP (left) and P (right) coupling. The bright and dark areas in the nanomagnet regions in the MFM images correspond to ↑ and ↓ magnetization, respectively. All the scale bars are 1 μm.

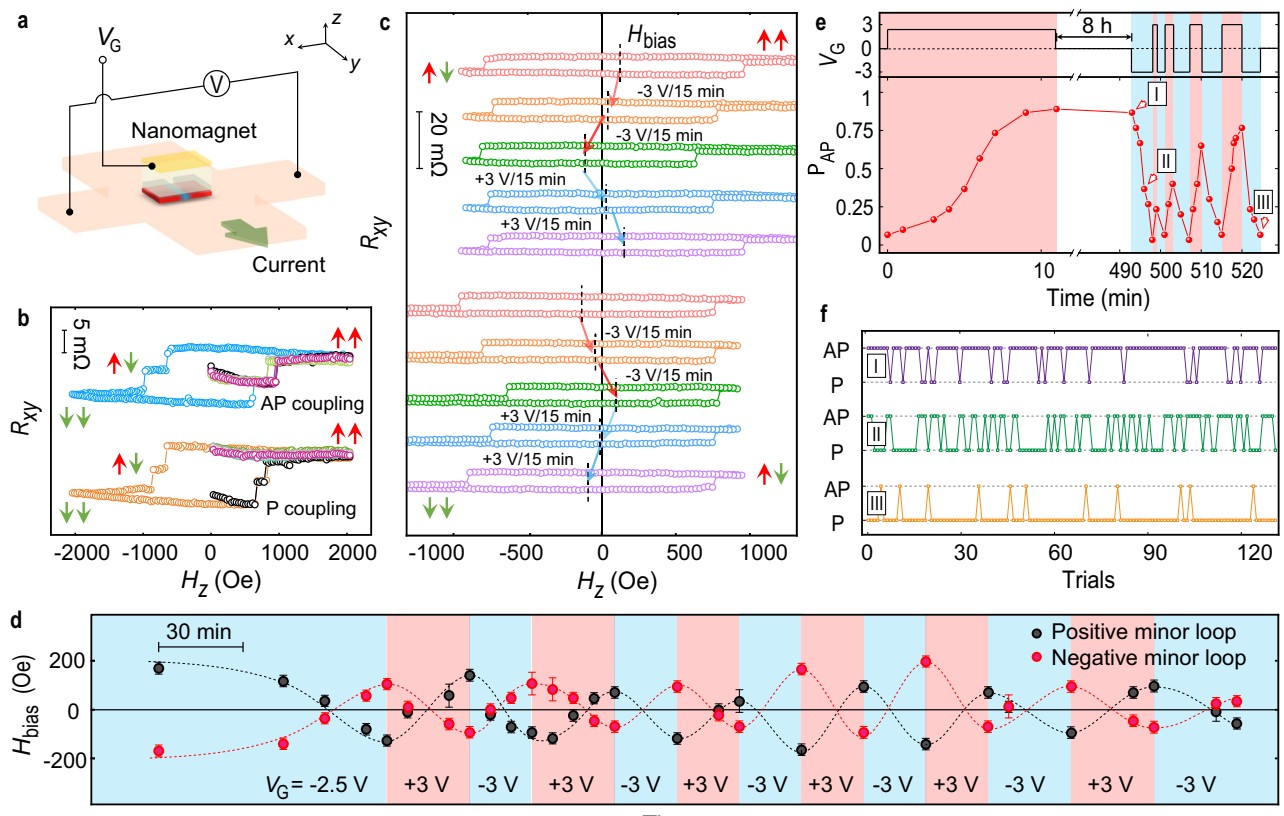

**Fig. 2 | Reversible conversion of AP/P magnetic coupling. a** Schematic of the Hall device used for electrical transport measurements. **b** Magnetic hysteresis loops of a nanomagnet element on the Hall bar with AP (top) and P (bottom) coupling. The full hysteresis loops show three resistance levels corresponding to magnetic configurations of ↑↑, ↑↓ (↓↑) and ↓↓. The half hysteresis loops are measured after employing the demagnetization protocol and with the magnetic field starting from zero. **c** Minor hysteresis loops with magnetic field starting from saturation. The magnitude of $H_{bias}$ is indicated with the black dashed lines. The $H_{bias}$ obtained from positive minor hysteresis loops (top 5 curves) has an opposite sign to that obtained from negative minor hysteresis loops (bottom 5 curves). **d** Evolution of $H_{bias}$ obtained from positive and negative minor hysteresis loops with respect to the gate voltage. The error bars represent the uncertainty in the estimation of $H_{bias}$. Red and black dashed lines are guides to the eye. **e** Percentage of AP magnetic configurations obtained after demagnetization (lower panel) with the gate voltage given in the upper panel. Each percentage is determined from 30 trials carried out on one device. Red- and blue-shaded regions highlight the application of positive and negative gate voltages, respectively. **f** Stochastic behavior of the demagnetized configuration for different coupling strengths. The labels I, II, and III correspond to the states indicated in **e**.

is demonstrated with the demagnetized configurations given in Fig. 1c. This coupling can be described as an effective exchange energy

$$E = -J(\mathbf{S}_1 \cdot \mathbf{S}_2) \qquad (1)$$

where $\mathbf{S}_1$ and $\mathbf{S}_2$ represent the direction of the adjacent Ising macrospins, which can only point ↑ or ↓, and $J$ is the coupling strength that can be tuned between P ($J > 0$) and AP ($J < 0$).

In order to systematically study the voltage-controlled coupling, nanomagnet elements were fabricated on a Hall bar for the electrical detection of the OOP magnetization via the anomalous Hall effect (Fig. 2a). Due to the large difference in size between the protected and the gate regions, the Hall resistance mainly reflects the state of the protected regions. Following the demagnetization protocol, we recorded hysteresis loops, which consistently show the switching between three magnetization levels, corresponding to ↑↑ (↓↓) at large positive (negative) fields and ↑↓ or ↓↑ at intermediate fields (Fig. 2b). Note that the Hall resistance starts at the middle resistance level corresponding to ↑↓ or ↓↑, indicating that the demagnetized magnetic configuration is AP. We then record minor loops to quantify the coupling strength between two protected regions. The minor loops are shifted horizontally by the exchange bias field $H_{bias} = 127 \pm 16$ Oe ($-130 \pm 16$ Oe) when the loop starts from positive (negative) saturation fields, confirming the presence of the AP coupling (red loops in Fig. 2c). The AP coupling

strength can be estimated from $J = H_{bias}MV_{OOP} = -2.5 \pm 0.3$ eV, where $M$ and $V_{OOP}$ are the magnetization and volume of the magnetic material in the protected region, respectively. After applying a negative gate voltage, the magnitude of $H_{bias}$ gradually decreases and eventually its sign is inverted, indicating the conversion from AP to P coupling (orange and green loops in Fig. 2c). The magnitude of exchange bias with P coupling is similar to that of AP coupling, and the strength is estimated to be $2.2 \pm 0.3$ eV. We could then verify that the demagnetized magnetic configuration is P since the Hall resistance started from either the highest or the lowest resistance levels corresponding to ↑↑ or ↓↓ (Fig. 2b), confirming the AP-to-P coupling conversion. Additionally, the magnetic coupling can be reversibly converted between AP and P by changing the $V_G$ polarity. As shown in Fig. 2d, we altered the $V_G$ polarity back and forth 13 times and $H_{bias}$ changed signs accordingly. Note that the time for the first conversion (~90 min) is longer than the subsequent conversion times (~30 min), which is also reflected by the time taken to go from the green to the purple loops in Fig. 2c. This could be related to the additional energy cost of the first detachment of the ions from their original positions as well as to the formation of ionic conduction paths that provide subsequent faster conversion[37].

When the coupling $J$ is close to zero, the magnetic configuration obtained after different demagnetization cycles exhibits a stochastic behavior. As shown in Fig. 2e, we start from the P state where the

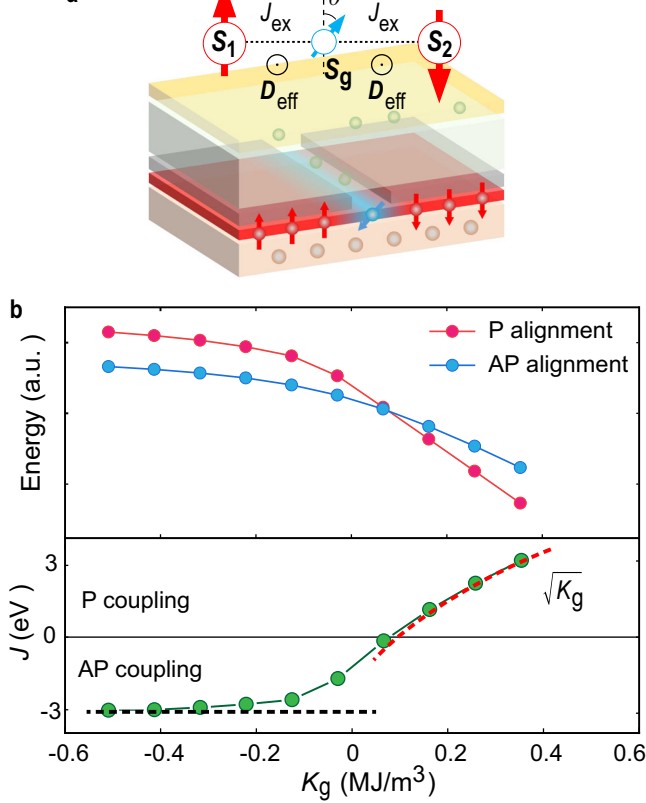

**Fig. 3 | Macrospin model of AP/P coupling conversion and micromagnetic simulations of a nanomagnet element. a** Schematic of macrospin model consisting of three macrospins representing the magnetizations in two protected regions and one gated region. **b** Total energy for the AP (blue) and P (red) alignment as a function of $K_g$ determined from the micromagnetic simulations. The difference between the AP and P energies gives the coupling strength $J$ (green). The AP coupling strength saturates at $-3.0$ eV, as indicated by the horizontal black dashed line, whereas the strength of P coupling increases with $\sqrt{K_g}$, as indicated by the red dashed line (see also Supplementary Information S3).

percentage of obtaining AP alignment, $P_{AP}$, is close to 0. When a gate voltage of $V_G = 2.5$ V is applied, $P_{AP}$ gradually increases over time and approaches 1. The percentage of AP alignment follows the Boltzmann law and is given by:

$$P_{AP} = \frac{e^{-J/k_B T_{eff}}}{e^{-J/k_B T_{eff}} + e^{J/k_B T_{eff}}} \quad (2)$$

where $k_B$ is the Boltzmann constant and $T_{eff}$ is the effective temperature resulting from the demagnetization protocol (Fig. S3). $P_{AP}$ is then regulated by $V_G$ that modifies the coupling strength, giving $P_{AP} > 0.5$ for $J < 0$ and $P_{AP} < 0.5$ for $J > 0$. We note that $P_{AP}$ remains almost unchanged after disconnecting the device for 8 h, indicating that the coupling is non-volatile thanks to the ionic gating effect[8]. We then apply a $V_G$ alternating between $\pm 3$ V and observe a synaptic plasticity of $P_{AP}$ that varies according to $V_G$ (Fig. 2e). We repeat the demagnetization process 130 times at $J < 0$, $J \approx 0$ and $J > 0$ (at I, II and III in Fig. 2e), and the magnetic configuration alternates randomly between AP and P with different percentages (Fig. 2f), which enables a modulation of the correlation between neighboring macrospins as required for Ising-type probabilistic computing[38–41].

## Mechanism for the AP/P coupling conversion
To provide deeper insight into the mechanism of the AP/P coupling conversion, we consider a macrospin model and calculate the coupling

strength for a given magnetic anisotropy in the gated region. As shown in Fig. 3a, $S_1$ and $S_2$ represent the direction of the magnetization in the protected regions, while $S_g$ represents the direction of the magnetization in the gated region. The total energy of this system is given by the sum of the symmetric exchange energy ($E_{ex}$), antisymmetric exchange energy ($E_{DM}$) and the magnetic anisotropy energy ($E_{an}$) of the gated region, which can be written as:

$$E = E_{ex} + E_{DM} + E_{an} = -J_{ex}\sum_{<i,j>} \mathbf{S}_i \cdot \mathbf{S}_j - \mathbf{D}_{eff}\sum_{<i,j>} \mathbf{S}_i \times \mathbf{S}_j - K_g V_g S_{gz}^2 \quad (3)$$

where $J_{ex}$ and $\mathbf{D}_{eff}$ denote the effective exchange energy and the DMI vector. $J_{ex} > 0$ and $\mathbf{D}_{eff} < 0$ in Pt/Co with left-handed chirality. $<i,j>$ represents all the possible combinations of the nearest-neighbor pairs of macrospins. $V_g$ is the volume of the magnetic material in the gated region and $K_g$ is the effective anisotropy constant of the gated region, which is experimentally tuned between IP ($K_g < 0$) and OOP ($K_g > 0$) by applying a gate voltage. The energies for the AP and P configurations are then:

$$E_{AP} = \begin{cases} 2D_{eff}; & \text{when } K_g V_g < -D_{eff} \\ -\dfrac{D_{eff}^2}{K_g V_g} - K_g V_g; & \text{when } K_g V_g \geq -D_{eff} \end{cases} \quad (4)$$

and

$$E_P = \begin{cases} \dfrac{J_{ex}^2}{K_g V_g}; & \text{when } K_g V_g < -J_{ex} \\ -2J_{ex} - K_g V_g; & \text{when } K_g V_g \geq -J_{ex} \end{cases} \quad (5)$$

The $S_1$-$S_2$ coupling strength is then determined from the difference between the energies $E_{AP}$ and $E_P$ and is given by: $J = (E_{AP} - E_P)/2$. On increasing $K_g$, $J$ increases from negative to positive, reflecting the AP-to-P coupling conversion (Fig. S5). Notably, if the gated region is strongly IP ($K_g \ll 0$), $J \approx D_{eff} < 0$, whereas if the gated region is strongly OOP ($K_g \gg 0$), $J \approx J_{ex} > 0$, so providing an intuitive picture for the AP/P coupling conversion resulting from the VCMA-mediated competition between the symmetric and antisymmetric exchange interaction.

In order to quantify the coupling strength, we carried out micromagnetic simulations using the MuMax3 code[42]. The initial magnetization in the protected regions is set to be ↑↓ (or ↑↑) with the cell magnetizations in the gated region pointing in random directions. The system is then allowed to relaxed until a stable magnetic state is reached. In the ↑↓ configuration, we find that a Néel domain wall separates the two protected regions, whose width decreases with increasing $K_g$. In the ↑↑ configuration, two 90° domain walls form, which eventually vanish as $K_g$ is increased (Fig. S7). The total energy as a function of $K_g$ is shown in Fig. 3b. The main features of the $E_{AP}$ ($E_P$)-$K_g$ and $J$-$K_g$ curves agree with those of the macrospin model (Fig. S5). When $K_g < 0$, $E_{AP} < E_P$ and $J$ saturates at $-3.0$ eV. When $K_g > 0$, $E_{AP} > E_P$ and $J$ increases with $\sqrt{K_g}$ (Supplementary Information S3). The experimentally obtained coupling strength ($-2.5$ eV for AP coupling and 2.2 eV for P coupling) are also in good agreement with the values estimated from the micromagnetic model (see Supplementary Information S3).

## Voltage-controlled phase transition in Ising artificial spin ices
As the magnetization in the protected regions can only point ↑ or ↓, it is possible to create an array of nanomagnet elements that mimic artificial Ising systems and display magnetic phase transitions that occur on changing the coupling strength. We first construct a one-dimensional (1D) Ising chain of nanomagnets by repeating an alternating structure of protected and gated regions in a line (Fig. 4a). The AP coupling in the as-fabricated Ising chain leads to

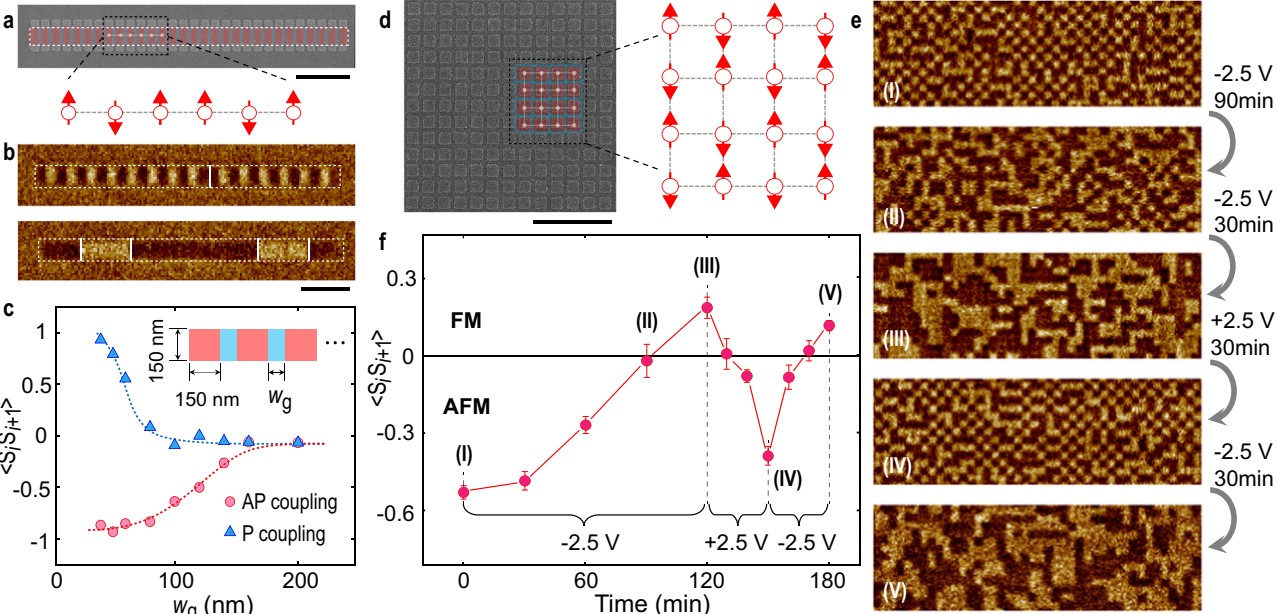

**Fig. 4 | Voltage-controlled magnetic phase transition. a** Colored SEM image (top) and corresponding schematic (bottom) of a 1D Ising-like chain structure. The widths of the protected and gated regions are 150 nm and 50 nm. **b** MFM images of the chain element with AP (top) and P (bottom) coupling. **c** Nearest-neighbor correlation function $<S_iS_{i+1}>$ of AP and P coupling in a chain of 30 coupled regions as a function of gate width $w_g$. The dimensions of the gated region are indicated in the inset. The red and blue dashed lines are guides to the eye. **d** SEM image of the 2D Ising-like square lattice structure (left). Part of the image is indicated in color with a corresponding schematic to the right. **e** MFM image sequence of the 2D Ising-like square lattice showing reversible magnetic phase transitions between AFM and FM order. **f** $<S_iS_{i+1}>$ as a function of time on applying different gate voltages to in the 2D square lattice. The states corresponding to the MFM image sequence (**I** to **V**) are indicated. The error bars represent the standard deviation of $<S_iS_{i+1}>$ evaluated from five lattices with 15 × 15 nanomagnets. The bright and dark areas in the nanomagnet regions in the MFM images correspond to ↑ and ↓ magnetization, respectively. Red- and blue-shaded regions in the SEM images indicate the protected and gated regions. All the scale bars are 1 μm.

antiferromagnetic (AFM) order on demagnetization (Fig. 4b upper panel). After applying $V_G = -2.5$ V to the chain for 90 min, all of the couplings are converted to P and the chain of nanomagnets exhibits ferromagnetic (FM) ordering on demagnetization (Fig. 4b lower panel). We also vary the width of the gated region $w_g$ and find that the degree of both AFM and FM orders gradually decreases with increasing $w_g$ (Fig. 4c). Here the degree of the magnetic order is evaluated by determining the nearest-neighbor correlation function:

$$<S_iS_{i+1}> = \frac{\sum_{<i,j>} \mathbf{S}_i \cdot \mathbf{S}_j}{N} \qquad (6)$$

where $N$ represents the number of nearest-neighbor pairs of macrospins and the sum runs over all nearest-neighbor pairs. $<S_iS_{i+1}>$ is equal to 1 and −1 for perfect AFM and FM order in the 1D Ising chain, respectively.

We then construct a 2D Ising artificial spin ice (Fig. 4d). The demagnetized square lattice exhibits an AFM checkerboard pattern in the as-fabricated state (Fig. 4e; Panel (I))[20]. A phase transition from AFM to FM order can then be achieved by applying a negative gate voltage (Fig. 4e; Panels (II) and (III)). This transition can again be quantified by calculating the nearest-neighbor correlation function $<S_iS_{i+1}>$. By changing the $V_G$ polarity, the magnetic order is switched between AFM and FM with $<S_iS_{i+1}>$ varying between positive and negative values (Fig. 4f and Fig. S9). The $<S_iS_{i+1}>$ for FM order is smaller than that for AFM order, implying that the P coupling is weaker than the AP coupling. This could be due to the fact that the dipolar interaction becomes considerable in extended lattices and inhibits the formation of the FM order (Supplementary Information S7).

## Programmable Ising network and its application to neuromorphic computing

Taking advantage of the flexibility of voltage control, we can adjust the individual magnetic couplings in an Ising network independently, thus providing addressable control of the couplings and creating an electrically programmable array of coupled nanomagnets. To demonstrate this feature, we fabricated a four-spin chain element where every gated region has a dedicated electrode and can be controlled independently (Fig. 5a, b). Analogous to the encoding method used in a binary system, we define the AP coupling as binary "1" ($J < 0$) and P coupling as binary "0" ($J > 0$). This means that the positive and negative gate voltages are defined as binary "1" ($V_G = 2.5$ V) and "0" ($V_G = -2.5$ V), respectively. The four-spin chain element has $2^3 = 8$ coupling configurations ($J_1, J_2, J_3$) corresponding to "000", "001", "010", "011", "100", "101", "110" and "111", which can be programmed one-by-one by individually setting the three gate voltages ($V_1, V_2, V_3$) (Fig. 5c). For instance, if the gate voltages are all positive i.e., ($V_1, V_2, V_3$) = "111" in terms of the electric signal, all the couplings are AP i.e., ($J_1, J_2, J_3$) = "111" in terms of the coupling configuration. In this case, the demagnetized magnetic configuration has a high percentage of AP alignment for all nearest-neighbor pairs $\mathbf{S}_1|\mathbf{S}_2$, $\mathbf{S}_2|\mathbf{S}_3$ and $\mathbf{S}_3|\mathbf{S}_4$ (Fig. 5d). If $V_1$ is switched to negative then ($V_1, V_2, V_3$) = "011" and ($J_1, J_2, J_3$) = "011". Therefore, after demagnetization, the $\mathbf{S}_1|\mathbf{S}_2$ pair exhibits P alignments with a high percentage, while the $\mathbf{S}_2|\mathbf{S}_3$ and $\mathbf{S}_3|\mathbf{S}_4$ pairs remain AP aligned with a high percentage (Fig. 5e). In this manner, we can obtain all eight coupling configurations (Fig. S11).

By extending this scheme to a more complex 2D network, it is feasible to construct a programmable Ising network whose couplings can be electrically adjusted between AP and P. Such a network has applications for both Boolean and non-Boolean computing. In the Supplementary Information S11, we describe the realization of reconfigurable Boolean logic gates such as a controlled-NOT and a controlled-Majority gate (Fig. S14), in which a dual logic functionality

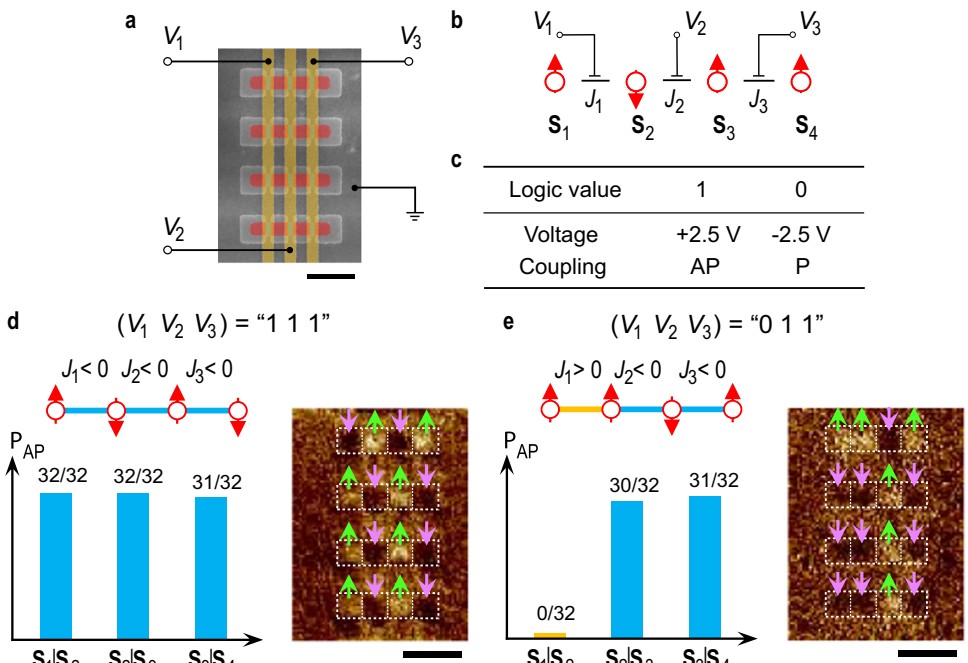

**Fig. 5 | Addressable control of magnetic coupling in a four-spin chain.**
**a, b** Colored SEM image and corresponding schematic of a programmable four-spin Ising chain. Red- and blue-shaded regions indicate the protected and gated regions, and yellow-shaded regions indicate the gate electrodes. **c,** Programming rules for the gate voltage and coupling. **d, e** Coupling configurations for "111" **d** and "011" **e** programmed by applying the corresponding electric voltages. The blue and yellow connecting lines represent AP and P coupling, respectively. The percentages

of AP alignment for the spin pairs $S_1 | S_2$, $S_2 | S_3$ and $S_3 | S_4$ after demagnetization are shown (left), illustrating the programmed coupling configuration. Each percentage is obtained from the measurement of 32 elements. The MFM images of four element structures are shown with the bright and dark areas in the nanomagnet regions correspond to ↑ and ↓ magnetization, respectively, which is indicated with green and purple arrows (right). In order to guarantee the complete AP/P conversion, the gate voltages are applied for 90 min. All the scale bars are 500 nm.

can be implemented by controlling the polarity of the control voltage. Moreover, our approach offers an efficient way to build an Ising-type neural network whose vertices are magnetically coupled to each other and each coupling is electrically adjustable. In particular, many combinatorial optimization problems that are ubiquitous in fields such as artificial intelligence, bioinformatics, drug discovery, cryptography, logistics and route planning, can be mapped to an Ising network with specific Hamiltonians[43,44]. The solution of such problems can be obtained by finding the spin alignment that corresponds to the ground state of the Ising network. In spintronic-based neuromorphic computing schemes, however, the couplings between vertices in an Ising-type neural network are usually achieved with additional CMOS circuits or by resistive crossbar arrays[40,45,46]. To illustrate the capability of solving combinatorial optimization problems using a magnetically-coupled network, we experimentally solve a benchmarking Max-Cut problem in an eleven-spin Ising network (Fig. 6).

The Max-Cut problem is frequently used for circuit design and machine learning[47,48], and is one of the most basic combinatorial optimization problems. In a typical Max-Cut problem, one starts with a system (a graph) in which a certain number of elements (the vertices of a graph) are related to each other by pairwise connections (the edges of a graph) with assigned weights. Finding the solution to the Max-Cut problem consists of maximizing the total weight of the edges between two mutually exclusive subsets of vertices. In our implementation, the nanomagnets represent the vertices of the graph, which are separated into two sets according to their magnetization, ↑ or ↓. The coupling strengths $J_{ij}$ correspond to the weights of the edges $w_{ij}$ in the graph. Solving the Max-Cut problem is equivalent to minimizing the energy of an Ising network with the same connections. Since we can electrically program the strength of each individual coupling, our nanomagnetic system can serve as a combinatorial optimization problem solver. In Fig. 6a, b, an SEM

image and schematic of an eleven-spin Ising network is shown, which represents a specific Max-Cut problem where each nanomagnet has either two or three connections. As shown in Fig. 6c (upper panel), the demagnetized configuration of the Ising network whose couplings are all AP ($J_{ij} < 0$) reveals the solution for the Max-Cut problem of a graph whose edge weights are all positive ($w_{ij} > 0$). In order to change the weight $w_{34}$ to negative ($w_{34} < 0$), the corresponding coupling $J_{34}$ is tuned to P ($J_{34} > 0$) by applying $V_G = -2.5$ V for 90 min and the demagnetized magnetic configuration gives the new solution (Fig. 6c, lower panel). Therefore, the solution of Max-Cut problems can be obtained by relaxing our physical system to its ground state within a finite time. Different Max-Cut geometries can be implemented using our approach (Fig. S12). However, due to the geometric limitations of a 2D nanomagnetic network and the short-range nature of chiral coupling, only nearest-neighbor connections are possible. More general network structures can be implemented by exploiting the long-range dipolar interaction[23,49], or by electrically coupling distant elements using the spin-transfer torque effect[40] (Fig. S13). Moreover, the ability to electrically program the magnetic couplings permits the adjustment of the Max-Cut problem at run-time, so enabling hardware-level programmability of the solver.

## Discussion
We have shown that we can electrically tune the magnitude and sign of the lateral coupling between nanomagnets by taking advantage of the antisymmetric exchange interaction and modifying the magnetic anisotropy in a gate region between the nanomagnets. The change in magnetic anisotropy is a consequence of the electrochemical reaction localized at the interface and ion migration in the gate dielectric under an electric field. The time required for the modification is limited by the reaction rate and ionic mobility, and can be

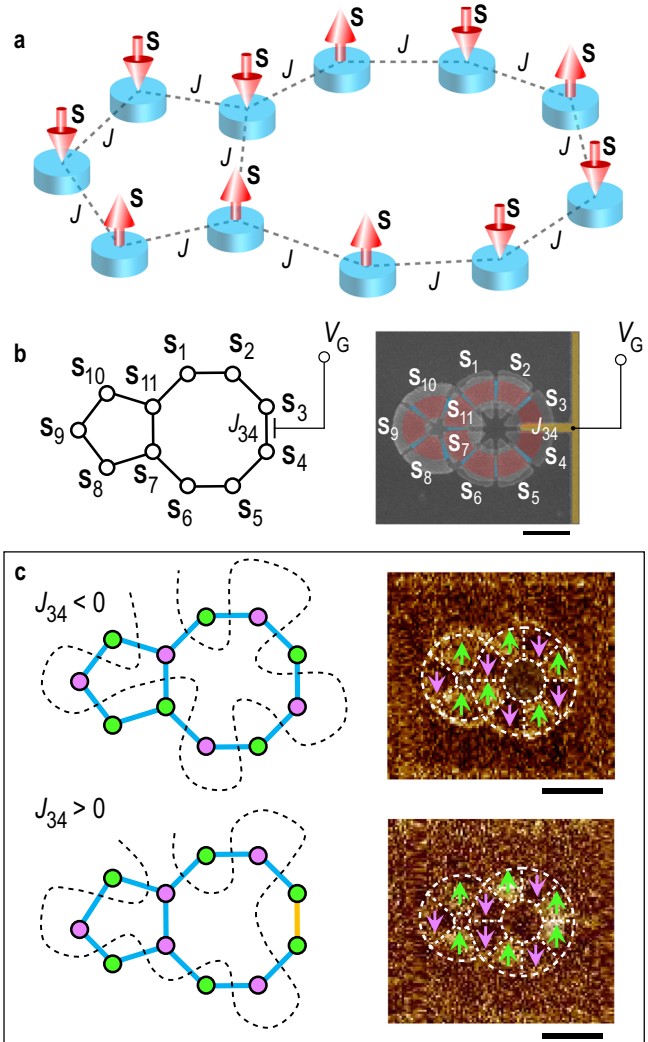

**Fig. 6 | Programmable Ising network as hardware solver for Max-Cut problems.**
**a** Schematic of a programmable Ising network based on coupled nanomagnets.
Each coupling strength can be programmed by applying the corresponding electric
voltage. **b** Colored SEM image and corresponding schematic of programmable
eleven-vertex Ising network. In the SEM image, red- and blue-shaded regions indi-
cate the protected and gated regions, while the yellow-shaded region indicates the
gate electrode. **c** Solutions to Max-Cut problem obtained from MFM images of
demagnetized devices for the cases when $J_{34}$ is programmed to be AP (top) and P
(bottom). The blue and yellow connecting lines in the schematics represent AP and
P coupling. The black dashed line in each of the schematics indicates the cut lines
separating vertices into two complementary sets (in green and purple), which is the
solution to the Max-Cut problem with the corresponding weights. The bright and
dark areas in the nanomagnet regions in the MFM images correspond to ↑ and ↓
magnetization, respectively, which is indicated with green and purple arrows. All
the scale bars are 500 nm.

reduced, in principle, using a high-mobility ion conductor[50] or the
electronic-version of the VCMA effect, which can be operated at GHz
frequencies[7].

Our approach offers the possibility to investigate collective phe-
nomena, such as the coupling-dependent phase diagram[51] and phase
boundaries of mixed FM/AFM Ising-like artificial spin ices[52,53], as well as
the exotic magnetic phase of a spin glass[18,20]. Moreover, we have pro-
vided proof-of-concept demonstrations of reprogrammable nano-
magnetic Boolean logic gates and combinatorial Ising solvers, which
will inspire future research on unconventional computing devices
based on nanomagnets.

## Methods

### Device fabrication

Films of Ta (5 nm)/Pt (5 nm)/Co (1.5 nm)/Al (2 nm) were deposited on a
200-nm-thick $SiN_x$ layer on a silicon substrate using d.c. magnetron
sputtering at a base pressure of $<2.7 \times 10^{-6}$ Pa and an Ar pressure of
0.4 Pa during deposition. The Al layer was oxidized to induce perpen-
dicular magnetic anisotropy in the Co layer using a low-power (30 W)
oxygen plasma at an oxygen pressure of 1.3 Pa. The fabrication of the
voltage-controlled coupled nanomagnets was carried out with electron-
beam lithography. Continuous magnetic films were milled into the
shape of the bottom electrodes with Ar ions through a negative resist
(ma-N2401) mask. The upper Co/AlOx layers were milled through a
positive resist [poly(methyl methacrylate), PMMA] mask to create the
nanomagnets and lattice structures. Using electron-beam evaporation,
a protective layer of Cr (2 nm)/SiO₂ (8 nm) was deposited, with the
protected region defined using a lift-off process through a second
PMMA mask patterned by electron-beam lithography. Then an elec-
trolyte layer of GdOₓ (30 nm) was deposited using reactive magnetron
sputtering at an Ar pressure of 0.4 Pa and with a mixed gas flow of
50 sccm Ar and 1 sccm O₂. In order to promote the ionic gating effect, a
short milling process was implemented to partially remove the AlOₓ
layer in the gated region. Finally, top electrodes of Cr (2 nm)/Au (3 nm)
were fabricated using electron-beam lithography combining electron-
beam evaporation with a lift-off process. The base pressure for the
electron-beam evaporation was $<1.3 \times 10^{-4}$ Pa and the deposition rate
for Cr, SiO₂ and Au was 0.5 Å/s. The main steps of the device fabrication
are shown in Fig. S1. The magnetic anisotropies in the protected and
gated regions were confirmed with polar MOKE measurements (Fig. S1).

### MFM measurements

The MFM measurements were performed with a Bruker Dimension Icon
Scanning Station mounted on a vibration- and sound-isolation table
using tips coated with CoCr. To minimize the influence of the stray field
from the MFM tip during the measurements, low-moment MFM tips
were adopted. We repeated the MFM measurements and found that the
magnetization in the nanomagnets remains unchanged, confirming that
the MFM tips do not alter the magnetic configurations. For the voltage-
control of the coupled nanomagnets, the samples were mounted on a
dedicated holder and connected to a source meter (Keithley 2400) with
wire bonding. The MFM images were captured after employing the
demagnetization protocol (Supplementary Information S2). All of the
MFM measurements were performed at room temperature and under
ambient conditions.

### Electrical measurements

For electrical measurements, the magnetic films were patterned onto a
1.5 μm-wide Hall cross using electron-beam lithography and the cou-
pled nanomagnet elements were located in the center of the Hall cross.
The devices were then connected to a source meter (Keithley 2400)
and voltmeter (Keithley 2182) with wire bonding. All of the electrical
measurements were performed at room temperature and under
ambient conditions.

### Micromagnetic simulations

To understand the mechanism of the AP/P coupling conversion,
micromagnetic simulations were carried out with the MuMax3 code[42]
using a computation box containing $1000 \times 1000 \times 1$ cells with
$2 \times 2 \times 1.5$ nm³ discretization and the following magnetic parameters:
saturation magnetization $M_S = 0.9$ MA m⁻¹, effective OOP anisotropy
field in the protected region $H_{eff} = 5$ kOe, exchange constant
$A = 16$ pJ m⁻¹ and interfacial DMI constant $D = -1.5$ mJ m⁻².

## Data availability

The data that support the findings of this study have been deposited in
the Zenodo database, at https://doi.org/10.5281/zenodo.7170022.

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

## Acknowledgements

We acknowledge funding from the National key research and development program of China (Nos. 2021YFB3501301, 2021YFB3503101 and 2022YFA1203904), EU FET-Open RIA project SpinENGINE (No. 861618), Swiss National Science Foundation (Grant Agreement 200021_182013 and 200020_200465) and National Natural Science Foundation of China (Nos. 52271160, 51731001, 11805006, 12241401, and 11975035).

## Author contributions

Z.Luo, L.J.H., and P.G. conceived the work and designed the experiments; A.H. and Z.Liu fabricated the devices. C.Y. and Z.Liang performed the MFM and electric measurements with the support of Y.F.; C.Y. and Z.Liang analyzed and interpreted the data with the help of M.H., Y.H., and J.Y.; C.Y. and L.W. performed the micromagnetic simulations; Z.Luo, P.G. and L.J.H. worked on the manuscript together. All authors contributed to the discussion of the results and the manuscript revision.

## Competing interests

The authors declare no competing interests.
