## [Peer Review File · Nature Communications]

Reviewers' Comments:

Reviewer #1:

Remarks to the Author:

The authors have made noteworthy contributions in their research paper. The reviewer expresses gratitude to the authors for addressing the raised concerns.

However, regarding the reviewer's earlier concern about the necessity of designing a complex layout for a relatively simple problem, such as the 11-node toy example presented in S7 and S11, the authors have not responded satisfactorily.

Although the cited papers offer examples of unique physical layouts corresponding to specific Ising networks, they address entirely different problems. Therefore, the explanation provided in those papers fails to illuminate the advantages of employing such a complex structure for the aforementioned 11-node toy example.

During the discussion on the geometric limitations of the 2D Ising network, the paper introduces a novel solution in the form of a Hybrid MTJ/Ising network structure. Supplementary Information S10 provides detailed insights into this proposed construction. However, it is unclear why the original proposal for the 2D Ising network is necessary if the Hybrid MTJ/Ising network structure can tackle the Max cut problem. In essence, the novelty and distinctive features of the 2D Ising network scheme require further clarification. Furthermore, the Hybrid Ising-like nanomagnetic network structure presented in the paper exhibits only a single electric coupling. It would be beneficial to incorporate multiple electrical couplings to enhance the credibility and efficacy of the proposed structure.

Reviewer #2:

Remarks to the Author:

I thank the authors for replying to my comments so carefully. With the addition of a description of the applications and limitations of the Ising machine they have realised I think the paper is suitable for Nature Comms. The strengths of the paper are many and I don't think the slow operation is a deal-breaker for a first demonstration paper.

Response to the Referees:

We are very grateful to the Referees for their further comments. In response to their criticism, we have revised the manuscript accordingly. We give a point-by-point response below with the original comments by the Referees reported in blue and our reply in black. A summary of key revisions is given at the end of this document.

Reviewer #1:

The authors have made noteworthy contributions in their research paper. The reviewer expresses gratitude to the authors for addressing the raised concerns.

However, regarding the reviewer's earlier concern about the necessity of designing a complex layout for a relatively simple problem, such as the 11-node toy example presented in S7 and S11, the authors have not responded satisfactorily. Although the cited papers offer examples of unique physical layouts corresponding to specific Ising networks, they address entirely different problems. Therefore, the explanation provided in those papers fails to illuminate the advantages of employing such a complex structure for the aforementioned 11-node toy example.

Reply:

We thank the Referee for their recognition of our contributions and that we have addressed the previous comments, as well as for the constructive comments to help us to further refine the manuscript.

The Referee is concerned about the necessity of fabricating a “complex” structure to solve a “simple” Max-Cut problem. Despite the fact that the 11-node Max-Cut problem appears to be simple, the conventional CMOS-based approach to solve this problem is very complicated. In order to find the solution to the Max-Cut problem, one should calculate the weight values for all possible spin configurations and choose the configuration with the maximum weight value. For an 11-node Max-Cut problem, the number of possible spin configurations is $2^{11} = 2048$ and, in order to perform this procedure, the CMOS-based hardware should contain multiple electronic circuits to execute the required operations such as selection, addition, multiplication, comparison and memory.

Every functional circuit is composed of tens of transistors and can only perform sequential operations. Even for the CMOS-based approach that mimics the annealing process with virtual spin vertices, each virtual spin vertex consists of a spin memory circuit, an exclusive OR (EOR) circuit and a majority-vote circuit, which is constructed using hundreds of transistors [T Takemoto, et al. IEEE Journal of Solid-State Circuits 55, 145-156 (2019)]. Moreover, the virtual annealing process needs additional control circuits to synchronize all the spin vertices.

In comparison, the principle behind the layout in our nanomagnetic Ising network is to directly map the graphic network onto a physical structure, which is relatively straightforward. Here the relative position of the nanomagnets simply corresponds to the arrangement of the vertices in the network and the shape of the nanomagnet corresponds to the coordination (or connection) number of the vertex. The advantages of our physically-coupled Ising network are the built-in reconfigurable magnetic coupling and the simplicity of the demagnetization protocol required to find the ground state. So, compared to CMOS-based hardware, our approach based on nanomagnetic Ising networks requires less electronic components and hence has smaller size.

Moreover, our approach can be scaled up to have a large number of spin vertices. This is because the increase of vertex number in the Ising network only extends the nanomagnetic structure, while the device fabrication process and demagnetization procedure to obtain the ground state remain the same. In contrast, for the conventional approach based on CMOS-based hardware, the required time and energy consumption to find the solution increase exponentially with the increase of spin vertex number, as it needs to run all the possible spin configurations with the total number of configurations given by 2^N where N is the spin vertex number. The total number of configurations becomes huge for a large number of spin vertices. For example, even for 15 spin vertices there are more than 30'000 configurations.

We have added a few descriptive sentences in the Supplementary Information to highlight the potential advantages of coupled nanomagnets compared to approaches based on CMOS circuits for solving combinatorial optimization problems. At the same time, we point out that benchmarking proof-of-principle demonstrations of novel computing hardware

against established CMOS technology, with decades of significant government and industry-sponsored R&D behind it, is a daunting task that is fraught with problems related, e.g., to defining a comparable set of tasks and overheads assigned to peripheral electronics. Additionally, compact models of the nanomagnetic devices would have to be developed and integrated within a process design kit, which is beyond the scope of our work.

During the discussion on the geometric limitations of the 2D Ising network, the paper introduces a novel solution in the form of a Hybrid MTJ/Ising network structure. Supplementary Information S10 provides detailed insights into this proposed construction. However, it is unclear why the original proposal for the 2D Ising network is necessary if the Hybrid MTJ/Ising network structure can tackle the Max cut problem. In essence, the novelty and distinctive features of the 2D Ising network scheme require further clarification.

Reply:

In this manuscript, we demonstrate the first electrically programmable nanomagnetic Ising network, which can be used to solve combinatorial optimization problems. Whereas we can exploit a nanomagnetic Ising network to map some combinatorial optimization problems to 2D Ising networks with only nearest-neighboring couplings, it is challenging to establish crossed connections beyond nearest neighbors due to the geometric limitation of the 2D physical structure.

In order to realize a more general network, we proposed the hybrid MTJ/Ising network in which the spin vertices can be electrically coupled via spin transfer torques in MTJs. Hence, the 2D Ising network is the basis for the hybrid structure.

We have added a paragraph in the Supplementary Information to clarify the difference between the 2D Ising network and hybrid MTJ/Ising network.

Furthermore, the Hybrid Ising-like nanomagnetic network structure presented in the paper exhibits only a single electric coupling. It would be beneficial to incorporate multiple electrical couplings to enhance the credibility and efficacy of the proposed structure.

Reply:

We thank the Referee for pointing out this additional possibility. In Supplementary Information S10, we conducted the simulation of a hybrid MTJ/Ising network with one electrical coupling. The simulation results verified the effectiveness of electrical coupling in the hybrid structure. Similarly, multiple electrical couplings can be incorporated to realize a more complex network, by adopting the methodology used for p-bit computation [W. A. Borders, et al. Nature 573, 390–393 (2019)]. In a general case, when the vertex S_i has N_i electrical couplings interacting with $S^1 \dots S^{N_i}$, the electric current I_i is given by:

$$I_i = \sum_{j=1}^{N_i} I_{ij} \text{sign}(S_j)$$

where I_{ij} is the intensity of the electric current corresponding to the coupling strength J_{ij} . During the demagnetization protocol, the magnetization of the spin vertex with virtual couplings is read via the MTJ resistance with a low electric current. Then the relatively large electric currents are calculated and injected into the corresponding MTJ, leading to the weighted electrical coupling.

We have added a paragraph in the Supplementary Information to clarify the implementation of multiple electrical couplings.

Reviewer #2:

I thank the authors for replying to my comments so carefully. With the addition of a description of the applications and limitations of the Ising machine they have realised I think the paper is suitable for Nature Comms. The strengths of the paper are many and I don't think the slow operation is a deal-breaker for a first demonstration paper.

Reply:

We thank the Referee for taking the time to review our manuscript and their support for publication in Nature Communications.

Summary of key revisions

In the Supplementary Information:

- New sentences on page 20 to clarify the difference between the 2D Ising network and hybrid MTJ/Ising network.
- New sentences on page 22 to clarify the implementation of multiple electrical couplings.
- New paragraphs on pages 22-23 to highlight the potential advantages of coupled nanomagnets.
- New reference 64 to support the potential advantages of coupled nanomagnets compared to approaches based on CMOS circuits for solving combinatorial optimization problems.